# Investigating Cardiorespiratory Interaction Using Ballistocardiography and Seismocardiography—A Narrative Review

**DOI:** 10.3390/s22239565

**Published:** 2022-12-06

**Authors:** Paniz Balali, Jeremy Rabineau, Amin Hossein, Cyril Tordeur, Olivier Debeir, Philippe van de Borne

**Affiliations:** 1Laboratoray of Physics and Physiology, Université Libre de Bruxelles, 1050 Brussels, Belgium; 2Laboratory of Image Synthesis and Analysis, Université Libre de Bruxelles, 1050 Brussels, Belgium; 3Department of Cardiology, Erasme Hospital, Université Libre de Bruxelles, 1050 Brussels, Belgium

**Keywords:** ballistocardiography, seismocardiography, respiratory signal, heart rate variability, sleep breathing disorders, sleep apnea, cardiopulmonary interaction, cardiorespiratory interaction, wearable cardiac monitoring

## Abstract

Ballistocardiography (BCG) and seismocardiography (SCG) are non-invasive techniques used to record the micromovements induced by cardiovascular activity at the body’s center of mass and on the chest, respectively. Since their inception, their potential for evaluating cardiovascular health has been studied. However, both BCG and SCG are impacted by respiration, leading to a periodic modulation of these signals. As a result, data processing algorithms have been developed to exclude the respiratory signals, or recording protocols have been designed to limit the respiratory bias. Reviewing the present status of the literature reveals an increasing interest in applying these techniques to extract respiratory information, as well as cardiac information. The possibility of simultaneous monitoring of respiratory and cardiovascular signals via BCG or SCG enables the monitoring of vital signs during activities that require considerable mental concentration, in extreme environments, or during sleep, where data acquisition must occur without introducing recording bias due to irritating monitoring equipment. This work aims to provide a theoretical and practical overview of cardiopulmonary interaction based on BCG and SCG signals. It covers the recent improvements in extracting respiratory signals, computing markers of the cardiorespiratory interaction with practical applications, and investigating sleep breathing disorders, as well as a comparison of different sensors used for these applications. According to the results of this review, recent studies have mainly concentrated on a few domains, especially sleep studies and heart rate variability computation. Even in those instances, the study population is not always large or diversified. Furthermore, BCG and SCG are prone to movement artifacts and are relatively subject dependent. However, the growing tendency toward artificial intelligence may help achieve a more accurate and efficient diagnosis. These encouraging results bring hope that, in the near future, such compact, lightweight BCG and SCG devices will offer a good proxy for the gold standard methods for assessing cardiorespiratory function, with the added benefit of being able to perform measurements in real-world situations, outside of the clinic, and thus decrease costs and time.

## 1. Introduction

The heart is one of the most vital organs in the body, acting as a pump to carry blood to all other parts [1]. The pumping movement is caused by a mix of electrical and mechanical activity, leading to blood circulation [1]. There are various methods for monitoring the activity of the heart [1]. Electrocardiography (ECG) is the most popular one and measures the electrical potential variations of muscle fibers in various areas of the heart [1]. It is measured on the body surface using electrodes whose number and position depend on the chosen configuration [2]. The ECG waveform in each heartbeat contains three complexes: P, Q, R, S, and T (see Figure 1) [1]. These complexes correspond to depolarization and repolarization events in the cardiac chambers [2]. The P wave represents atrial depolarization; the QRS wave represents ventricular depolarization; and the T wave represents ventricular repolarization [2].

Another modality that can be used to investigate the heart’s activity is phonocardiography (PCG), which records the sounds and murmurs produced by the heart during a cardiac cycle [1]. The mechanical activity of the heart can produce four separate sounds, known as S1, S2, S3, and S4. S1 and S2 are physiologic sounds, corresponding to the beginning of the ventricular systole and diastole, respectively. However, S3, S4, murmurs, and other noises frequently indicate a sickness or abnormalities [1]. S1 is usually a low-frequency and high-amplitude signal, while S2 has high frequency and low amplitude [1].

In addition to these techniques that are well known and widely used in the clinical setting, there are other modalities for cardiac monitoring. In particular, some are based on the mechanical activity of the heart, such as ballistocardiography (BCG) and seismocardiography (SCG).

BCG was discovered in 1877 by Gordon [3]. He reported his technique for graphically capturing the motions of the whole body caused by blood flow ejection at each cardiac contraction [3]. Henderson [4], Heald and Tucker [5], and several other researchers followed up with reports on their research into this subject with regard to breathing effects on the signal. However, the contemporary clinical age of BCG did not begin until 1936, after the comprehensive work by Starr [6,7].

The BCG signal can be measured as a displacement, velocity, acceleration, or force signal [8]. It includes movement along all three axes [8]. The initial BCG systems, however, focused on longitudinal (head-to-foot) BCG measurements, which were assumed to represent the largest projection of the 3D forces induced by cardiac ejection [8]. The peaks and valleys in the longitudinal BCG waveform are represented in Figure 1. These BCG waves can be classified as pre-systolic (F, G), systolic (H, I, J, K), and diastolic (L, M, N). The BCG record of a healthy subject is reported in Figure 1, with dominant H, I, J, K, and L waves forming a W-shaped wave and smaller so-called diastolic components [9].

SCG was introduced by Bozhenko in 1961 [10]. It consists of measuring the micro-vibrations caused by contractions of the heart at the chest surface. SCG components can be presented in all three axes of a tri-axial accelerometer, each presenting a distinct pattern [8]. However, the bulk of SCG research in the literature only looks at the amplitude of the dorsoventral component [8]. During various phases of the cardiac cycle, the variations in cardiac contraction forces result in several peaks and valleys in the SCG signal, corresponding to distinct cardiovascular events, referred to as fiducial points [11]. These reference points can be measured with the same accuracy as the equivalent ECG parameters [11]. It is shown that these points correspond to: the flow through the mitral valve during the left ventricle’s early rapid filling phase; the atrial systole; the ventricle isovolumic contraction; the mitral valve closure; etc. (see Figure 2). Typically, the timing of the largest SCG wave refers to the maximum flow through the aortic valve during fast ventricular ejection [11,12].

There are several technologies that are used for recording local vibrations on the chest wall, which are, in essence, similar to SCG. Forcecardiography [13], for instance, is a method for measuring the local forces imposed on the chest wall by the heart’s mechanical activity. The signal can be divided into three components: low-frequency, high-frequency, and heart sound. The high-frequency component shares many similarities with the SCG. However, as reported by Centracchio et al., it has not yet been demonstrated to provide the timings of SCG fiducial points with reasonable accuracy [14].

Another technology for recording chest wall vibrations is gyrocardiography [15], which measures heart-induced micro-vibrations using a gyroscope sensor mounted on the sternum. Thus, it can be considered a rotational SCG [15]. Vibrational cardiography [16] and kinocardiography [17] are also the other modalities used to detect vibrations via accelerations and/or angular velocities. The former is defined as a combination of SCG and gyrocardiography, and the latter is described as a combination of BCG and SCG in both linear and rotational dimensions.

All these cardiac signals are affected by respiration. Spontaneous pulmonary ventilation causes cyclical changes in pleural, alveolar, lung transmural, and intra-abdominal pressures [18]. Those periodic changes have different effects on the cardiovascular system and are the source of respiratory cardiovascular coupling [19]. First, the pulmonary stretch receptors stimulated during inspiration activate the Hering–Breuer reflex, provoking a positive chronotropic effect via respiratory-related inhibition of cardiac vagal motoneurons in the nucleus ambiguous [19,20]. Second, the thoraco-abdominal respiratory pump produces oscillations in venous return and left ventricular afterload, generating arterial pressure oscillations [20,21,22]. In turn, those latter oscillations trigger the baroreflex, causing a negative chronotropic effect that occurs in expiration [20,21,22]. During inspiration, the baroreflex inputs in cardiac vagal motoneurons are gated through central mediation or lung inflation afferents [23]. Third, a direct interaction occurs between the brainstem respiratory oscillator, which includes inputs from the pre-Bötzinger inspiratory neurons, and cardiac vagal preganglionic neurons in the nucleus ambiguous [19]. The cardiac vagal preganglionic neurons display a peak in discharge during post-inspiration, at the onset of expiration, and at the beginning of the reduction in heart rate [19].

This oscillation of the heart rate during respiration is called respiratory sinus arrhythmia (RSA). It involves heart rate variability (HRV) in synchrony with respiration, wherein the RR interval is reduced during inspiration and prolonged during expiration [24]. RSA is one of the categories of the so-called cardiorespiratory interactions. These cardiorespiratory interactions are indeed classified into three categories: RSA, which is defined by the variability in heart rate in the frequency of breathing; cardioventilatory coupling, which is characterized by the synchronization between the heartbeat and the onset of inspiration; and respiratory stroke volume synchronization, which is characterized by a constant phase difference between the right and left stroke volumes over a respiratory cycle [25].

While the impact of respiration on ECG [26] has already been extensively studied, this is much less the case for BCG and SCG. Indeed, the existing literature in the field of BCG and SCG have briefly outlined their interest in the study of RSA and HRV (see Section 5.2), even though they have a potential for much more. According to the recent systematic review on BCG and SCG research in the field of health care performed by Han et al. [27], only a few percent of studies focused on respiratory and sleep monitoring. Additionally, as shown in Table 1, there has been no review in the literature for this specific purpose. This review is thus dedicated to the potential of BCG and SCG applications in the field of cardiorespiratory interactions.

It should be noted that some sections mainly focus on BCG while only briefly mentioning SCG. The reason for this is that BCG was introduced earlier and was studied more in the past, so the literature on BCG is extensive, particularly about cardiorespiratory variations. The objective of this review is to attract readers’ attention to groundbreaking research and potential uses of BCG and SCG respiratory variations (the comparison of the contribution of this review and other reviews in the field is mentioned in Table 1). As shown in this table, no other reviews focused on this specific topic. In this article (see Figure 3), the causes of respiratory variations in BCG and SCG signals are discussed first (Section 3); then, the sensors used in recent studies are briefly mentioned (Section 4), studies that investigated these variations for various purposes are represented (Section 5), and finally, some ideas and open questions are raised (Section 6).

## 2. Search Strategy

To write this review, PubMed, IEEEXplore, and Google Scholar were searched for the terms “respiratory”, “respiration”, “breathing”, “breath”, “inspiration”, “expiration”, “apnea”, “lung”, “heart rate variability”, with “ballistocardiogram”, “ballistocardiograph”, “ballistocardiography”, or “seismocardiogram”, or any variations thereof. The publications including these combinations in the title, keywords, or abstract were selected for further study. Following an evaluation of the reference section of the original search results, more resources were discovered. Additionally, articles with “gyrocardiography”, “forcecardiography”, “kinocardiography”, or “mechanocardiography” were considered due to similarities or overlaps in the concepts. No restrictions related to publication dates were included. Finally, a total of 199 items relevant to the research were considered.

To be considered for inclusion in this review, a paper must be written in English and contain information about respiratory signals, HRV, or one of their applications. We excluded articles that did not contain any mention or description of respiratory signals extracted from BCG or SCG or simultaneously collected by means of other technologies unless it was deemed necessary for better comprehension.

## 3. The Cause of Respiratory Changes in BCG and SCG

### 3.1. BCG

In the early studies, it quickly became obvious that respiration had a significant influence on the shape of the BCG signal [4,44]. Since the effects of the respiratory cycle were so significant on BCG recordings, breath holding was considered required [4]. Nevertheless, variations in how the breath is held have a significant effect on the BCG signal [45,46]; the BCG waves are much larger when it is held in the inspiratory phase with the glottis open than when it is held in the expiratory phase [46,47]. During normal breathing, and in a healthy subject, the height of the IJ stroke increases with inspiration and decreases with expiration (see Figure 4) [48]. This respiratory variation may be altered in the presence of an illness [49].

There are different theories about the cause of respiratory variability on BCG records. De Lalla and Brown [48], for instance, suggested that an increased respiratory variability could very likely be the result of a reduced blood pool in the lungs. They considered this large pool of intravascular blood to function as a cushion between the right and left ventricles, preventing the left ventricle from transmitting the relatively severe changes in right ventricular output associated with intrapleural pressure changes.

Dock [50] suggested that most of the variation in the IJ waves of the BCG could be due to changes in force transmission rather than changes in force generation. When the heart is most transversely positioned during expiration, this results in a higher lateral component of force during left ventricular ejection. This is especially true in older adults with a transverse heart posture and a long aortic arch. According to the research he carried out, a significant percentage of the inspiratory increase in IJ waves may be attributed to a tensing of the mediastinal fibers. He proposed that when the diaphragm descends during inspiration, the descending heart tightens its connection with the rest of the body, allowing forces arising from the heart to be better transmitted to the body frame and increasing the amplitude of the force BCG. This hypothesis was later rejected by Noordergraaf [51], who calculated that the differences of heart body coupling during respiration would not account for the respiratory variations seen in BCG. Starr and Friedland [47] then compared BCG taken during normal breathing, breathing through an obstruction, and artificial breathing with an air blast to determine whether it was the change in position of the heart’s axis during the respiratory cycle that was causing respiratory variation. If Dock’s theory was correct, substituting pressure breathing with normal breathing should have had no effect on the BCG respiratory variation. They concluded that the variations are caused by changes of cardiac filling rather than position of the heart.

In 1966, Talbot et al. [52] assessed and found several diagnostically relevant BCG features using age-matched healthy and angina patients groups. It turned out that the inspiratory phase best discriminated coronary ischemia from control subjects. They further discussed the reason for BCG complexes in full inspiration providing the finest coronary heart disease information. They believed that Dock’s [50] “mediastinum” hypothesis does not explain why, even at mid-inspiration and in younger individuals, the diagnostic force differences are not conveyed. Furthermore, they believed that the stretched mediastinum idea ignores the scientific truth that a spring does not have to be rigid, tensed, or linear to transmit forces. On the other hand, they discussed the fact that the significant inspiratory rise in the right heart stroke volume that occurs together with a reduction in the left heart stroke volume is inadequate to account for the substantial increase in IJ wave height. Still, early systole may exhibit various velocity and acceleration patterns for the same stroke volume. Instead, they proposed another mechanism for the inspiratory increase in BCG force and its derivatives, as well as for its sensitivity to coronary heart disease. They indicated that the most diagnostic BCG features could be directly linked to the internal impedance of the myocardium. Their finding showed that the J amplitude reflects the myocardial limitation on initial velocity and its deterioration in coronary heart disease [52].

Finally, Prisk et al. also hypothesized that some changes in BCG waves during different respiratory phases may also be due to the changes in lung volume [53]. At lower lung volumes, the lung becomes more compliant and acts as a better shock absorber, cushioning the body from the heart’s movements.

### 3.2. SCG

The SCG signal amplitude and shape are also influenced by respiration [12,45,54,55]. Systolic time intervals, measured by SCG, which offer a quantitative estimate of left ventricular function, are changed depending on when they occur during the respiratory cycle, with notable changes between the inhale and exhale phases [56].

One approach to explain the respiratory variations in the SCG signal is to investigate the cause of the changes in heart sound [54]. It was demonstrated that sternal SCG signals can be decomposed into low-frequency cardiac output components (due to cardiac systole) and higher frequency heart sound components (due to valve closure) [12,57]. Respiration affects the amplitude and power of the two main heart sounds—S1 and S2—as well as the ratio of their powers. The modulation of the amplitude and power is in part caused by the change in the distance between the sound source and the accelerometer caused by respiration. Additionally, the changes in intra-thoracic pressure caused by respiration alter the ventricular preload and afterload, which can consequently lead to changes in the pressure gradients across the heart valves [54]. The series of changes in pleural pressure, venous return, right and left ventricular preload and afterload affect myocardial contraction force. In particular, they diminish the intensity of the S1 heart sound during inspiration, while the S2 heart sound becomes more intense (see Figure 4) [54]. It is also possible that changes in the air volume in the lungs impact the transmission of the vibrations and thus affect the SCG waveform.

### 3.3. Effect of the Modulation of Cardiac Output and Stroke Volume Caused by Breathing

Breathing has an impact on stroke volume, which can impact the BCG signal. Indeed, the IJ wave height in BCG records provides some indications of stroke volume [58]. In 1939, Star et al. [59] studied the link between cardiac output and BCG waves and provided a formula to compute stroke volume from the BCG signal (see Equation (1)). The results of this computation were consistent with experimental measurements of stroke volume using ethyl iodide.
(1)$$\mathrm{SV}=\kappa \sqrt{\left(2\int I+\int J\right){\;A\;C}^{1/2}},$$
where SV is the stroke volume in cm^3^; A is the aortic cross section in cm^2^; C is the length of one heart cycle in seconds; ∫I is the integral of the I wave in mm·s^−1^; ∫J is the integral of the J wave in mm·s^−1^; and κ is a proportionality factor of 33 cm^2^·mm^−1/2^·s^1/4^ given by Starr [59].

In 1951, Williams and Gropper [60] attempted to estimate the respiratory variations of right and left ventricles’ stroke volume using BCG and pressure-pulse methods, respectively. The experiments were conducted on normotensive and hypertensive subjects in normal breathing. The calculated right ventricular output was higher during inspiration, while the opposite variations were observed for the left ventricular output. Josenhans and van Brummelen [61] claimed that under normal conditions, and in case of a healthy subject, the sum of the IJ and JM wave heights was related to left ventricular stroke volume [62].

In 1945, Otis et al. [44] explored the influence of breathing on the cardiac output of subjects using BCG. Some of the experiments conducted in this study included pure oxygen inhalation, breathing against a high pressure during expiration, breathing against a high pressure during inspiration, and voluntary hyperventilation. They computed the stroke volume from the amplitudes of I and J waves using the Starr formula (1) and showed that these situations affect the heart rate and stroke volume. The mechanism underlying the stroke volume variation was further explored in Starr and Friedland’s study [47], showing that the right heart’s filling and, as a result, its output, immediately increases upon inspiration, but the left heart’s output does not increase for a few seconds. Immediately after expiration, the right heart’s output drops, followed shortly afterward by an equivalent fall in the left heart’s output. Finally, the right heart’s output fluctuates more than the left one’s during the breathing cycle.

Aside from these results, several authors postulated that the right ventricle had a stronger ballistic effect than the left ventricle. For instance, Baevsky et al. [63] hypothesized that lower diastolic pressure in the pulmonary arteries compared to the aorta could result in higher ejection velocity. Additionally, since the ballistic effect of contraction is directly proportional to the square of systolic ejection velocity, they concluded that it leads to a more significant ballistic effect. However, it was shown in numerical models by Starr and Noordergraaf [64] that the contribution of the right heart to the BCG signal is very small. Dock [50] also believed that the right ventricle played only a minor role in the generation of the BCG phenomenon and that the changes in its activity during respiration seemed insufficient to produce the respiratory variability in the BCG records. A recent numerical model by Rabineau et al. also confirmed this [65]. They also compared the stroke volumes predicted by Starr’s formula (1) with values generated by the computational model. They reported that although a positive correlation was found using this formula, it is not ideal for estimating intra-subject variations of stroke volume [65].

It is worth mentioning that although several studies have shown that parameters obtained from SCG can be highly correlated with stroke volume [66,67], to the writers’ knowledge, the respiratory variation of stroke volume and cardiac output in SCG is not discussed in the literature.

## 4. Sensors and Preprocessing

### 4.1. BCG and SCG Sensors

The only way to measure BCG in the early studies was to measure body displacement by a suspended bed that was free to move [3,4] or a table strongly coupled to its zero position [59]. In the last few decades, with the revival of the BCG technology [8], these large BCG sensors have been reduced in size to the point where they could be placed on or under a mattress [68,69,70,71,72], on a wheelchair [73], on the skin as a wearable device [17,74,75], or even record body movements in a contactless way [76] (see Table 2). Each of these approaches has its own advantages and can be chosen depending on the desired function.

For instance, wearable devices can collect data constantly during typical everyday activities and in any context, allowing for the assessment of a subject’s cardiovascular performance under diverse environmental conditions or stressors [8,77]. However, important limitations of such systems include motion artifacts and signal fluctuations with changes in the sensor’s location [30]. For example, chest-based signal is substantially different on the sternum compared to the lower left side of the chest [30]. They may also be more impacted by local movements of blood in vessels under the skin, and they often work with a battery that can limit the duration of experiments. Moreover, in order to obtain an accurate measurement, the sensor must be properly attached to the subject’s skin throughout the experiment, which can sometimes cause discomfort for the subject.

To evaluate sleep-related disorders (see Section 5.1.1) more comfortably than with traditional polysomnography, BCG has been shown to have a great potential, as it does not disrupt a subject’s normal sleep patterns to gather data [8]. Bed-based BCG can benefit from different kinds of sensors, from static charge sensitive beds [78,79,80] to pressure pads or pressure sensors [70,81,82], force sensors [69,83,84], loadcells in bed legs [85,86,87,88,89], fiber optic sensors [71,72,90], or electromagnetic film sensors [68,91]. These systems can offer comfort to the user, as well as the opportunity to easily install the systems at home [30]. However, there are some possible limitations, such as the postural effects on signal integrity and the resulting difficulties of matching the measured BCG signal with other physiological data, such as ECG [30]. Additionally, as the subject moves in the bed, the distribution of movement on the different axes will change.

Chair-based BCG is another form of BCG, which can be used in car seats to record the vital signs of drivers [92] or in wheelchairs [73]. These sensors, like bed-based devices, are comfortable and simple to use [30]. For these kinds of BCG recordings, electromagnetic film sensors are frequently used. The drawbacks of chair-based BCG may be the high influence of postural variations on the quality of the signal [8] and the unintended mixing of longitudinal and transverse BCG components [30].

**Table 2 sensors-22-09565-t002:** Summary of BCG sensors’ diversity in the articles discussed.

Sensor Type	Reference	Sensor Location
Suspended Bed	[3,4]	Bed
Load Cell	[85,87,89,93]	Under the legs of the bed
	[94]	Weighing scale
	[95]	Force plate
Force Sensor	[83,84]	Under mattress
Pressure Sensor	[70,82]	On mattress
Accelerometer	[74,75,96,97]	Wearable
Polyvinylidene Fluoride	[98]	Under mattress topper
Fiber Optic	[90]	Wearable
	[71]	Back of a chair
Optical Sensor	[72]	Under mattress
Fiber Bragg Grating	[99,100]	Wearable
Electro-Mechanical Films	[68,72,91]	On mattress
Static Charge Sensitive Bed	[78,79,80]	Bed
Radar	[76]	-

Similarly, SCG has been quantified using a variety of approaches, including wearable accelerometers [101,102], smartphones [103,104], and radars [105] (see Table 3). The specific position of the sensor on the recorded signal has an even greater effect on the SCG, since it is a measure of local vibrations [8]. However, as shown in Table 3, the sternum is the most frequently targeted area for wearable SCG sensors.

The benefits and drawbacks of using a wearable sensor for recording BCG also hold true for SCG. Furthermore, it has been demonstrated that utilizing a smartphone as an accelerometer can accurately monitor the resting heart rate [33]. However, if employed as a self-assessment tool, any changes in the acquisition protocol can alter data reliability, limiting the clinical acceptability of this method [33]. The pros and cons of radar SCG are exclusively discussed in Ref [36]. While utilizing a radar device reduces the sensor artifacts generated by connecting the sensor to the skin, it does have some drawbacks. Among them is the requirement that the subject’s chest remain in the sensor’s field of view, which severely limits its application in some fields, such as sleep monitoring [36].

As mentioned before, the BCG and SCG waveforms are under the influence of factors such as the position of the subject [8,36]. The reason is that the subject’s position can affect the blood flow and its distribution throughout the body. The direction of the recorded signal, which may depend on the subject’s position, is also important. Indeed, the BCG and SCG accelerations measured in the dorsoventral direction differ from those measured in the lateral or head-to-foot directions [8]. The recorded axes must be adjusted to fit the purpose of the study, or a three-axis accelerometer should be used [8].

In this regard, Vehkaoja et al. [98] evaluated the signals of five sensors and sensor placement combinations, including accelerometer and film-type force sensors made of piezoelectric polyvinylidene fluoride and electromechanical film material (EMFi) placed under the mattress topper and under the bedpost for monitoring heart rate and respiration. The signal amplitude was larger when placing the sensor under the chest area, but both BCG and respiration components were clear at almost all sensor locations. When the RMS values of each force sensor’s heartbeat and respiration components were compared in the position of lying on the right side, the sensors placed under the mattress topper near the thorax had a higher sensitivity to respiration movements than to heartbeats. The respiration component was especially weaker in the bedpost EMFi sensor.

In conclusion, when comparing the results of different studies, it is essential to consider the type of sensor, its exact placement, and the axes recorded. It is worth noting that some of the pioneer studies covered in this article are studied with different sensors than the ones used today. However, these meaningful results are very important and should be replicated using new sensors.

### 4.2. Preprocessing

As with other signals, the processing of BCG and SCG signals often entails preprocessing, feature extraction, and classification. These signals are prone to be contaminated by noise and artifacts. The sensors’ coupling to the body and motion artifacts are the primary causes of contamination of BCG and SCG signals [8,36]. The artifacts are usually removed using band-pass filters.

Respiration can also potentially contaminate the signal, primarily by generating a baseline wander caused by chest movement during respiration. It can also have an impact on the amplitude and time intervals [84,111], as already discussed. Nevertheless, these amplitude and interval fluctuations can sometimes be a source of information, for instance, for measuring the respiratory rate, phase, and amplitude. As a result, it can either be eliminated or included in the investigations, depending on the topic of interest.

To remove the baseline wander, respiratory and cardiac signals can be separated by band-pass filters [70,73,112,113] or moving average filters [114]. However, since the respiratory and cardiac components have overlapping spectra in the frequency domain, researchers sometimes select a more conservative cutoff frequency to assure complete removal of the respiratory component at the expense of information loss from the BCG and SCG component [115]. Even so, a number of studies have proposed more complex algorithms to preserve both components’ original waveforms. Yao et al. [116], for instance, presented an iterative locally projective noise reduction method, which separates signals based on the typical time and amplitude scale of deflections in the signals rather than their frequency components.

One approach to avoid integrating respiratory-induced differences in the analysis of BCG and SCG signals is to assign each heartbeat to a respiratory phase. Then, the heartbeats associated with different respiratory phases can be processed separately [45]. Tavakolian et al. [45] showed that the dissimilarities in morphology of the signal between the inspiratory and expiratory phases can decrease the accuracy of averaging. They used the respiratory information for improving the averaging and suggested that instead of removing the respiration effect from the signal, only the expiratory phases should be averaged. Indeed, the SCG signal is more consistent in expiratory beats than in inspiratory beats.

Depending on the data and application, further tasks, such as resampling [86], normalization [117], and segmentation [82], may be undertaken as well.

## 5. Studies That Combine Cardiac and Pulmonary Information

### 5.1. History of Diagnosis Based on Cardiorespiratory Variations of BCG

In some heart diseases, the variation of distinct waves with breathing phases may be altered. This can be used to diagnose certain diseases. The following subsections discuss the pathologies that were diagnosed or whose correlation with the respiratory variations of the signals was studied.

A case study on a female adult hospitalized for acute mixed congestive heart failure revealed that variations in heart and respiratory rate can be detected a few days prior to hospitalization [118]. Brown et al. [119] showed that the respiratory variations of BCG could be used to differentiate angina pectoris patients from other subjects. For this purpose, they defined a system of grading BCG records. Assuming that grade 0 represents normal records, the other four grades describe abnormalities. In grade 1, the IJ wave height is normal throughout the inspiratory phase. However, it decreases abnormally during the expiratory phase. In grade 2, half of the cycles are abnormal, and a drop in amplitude can also occur in the inspiratory phase, but still, it occurs mainly during expiration. In grade 3, the BCG complexes are still identifiable; however, the amplitude is low in both phases. In grade 4, the BCG amplitude is low, and the complexes are not identifiable. This classification is a good predictor of mortality, as shown in a long-term study by Starr et al. [120], in which they tracked the health status of subjects for more than 20 years to assess the prognostic value of BCG. They investigated the relationship between BCG amplitude and the development of heart disease and death. Those with normal BCG records outlived those with moderately abnormal and extremely abnormal records by over 4.5 years and 7.9 years, respectively.

Moreover, Anderson et al. [121] compared the respiratory variation of the amplitude of some BCG waves (H, I, J, K) in coronary artery disease, hypertension, and emphysema patients. Based on their findings, those with emphysema had a very large ratio between the inspiratory and expiratory amplitude of the I wave, while the coronary artery disease patients had the largest variation of the J wave. Furthermore, they defined a ratio between the amplitude of inspiratory IJ and expiratory IJ. They observed that coronary artery disease, hypertension, and emphysema patients all had an elevated IJ respiratory ratio, while this value for healthy subjects was reported to be less than 2. Brown and de Lalla [49] have established another measure called the respiratory variation index and computed as follows. The average of the largest inspiratory beats was used to compute the inspiratory minute volume. Similarly, the average of the smallest expiratory beats was used to compute the expiratory minute volume. Then, the difference between these two was divided by the body surface area of the participants. The result was below 450 for all normal subjects. In comparison, the respiratory variation index was higher than this threshold for all angina pectoris patients. Fernandez-Garcia [122] used Brown’s BCG classification to show that BCG can be used as a preoperative investigation in mitral stenosis patients. Since cardiac failure is more common in individuals with BCG grades 3 or 4, their findings indicate that the surgical risk is considerable, and post-operative care is critical.

Additionally, several researchers in the 1990s demonstrated that smoking increases the respiratory variation of BCG by decreasing the IJ wave during expiration [58,123,124]. It was hypothesized that muscular relaxation due to nicotine might explain this effect [58]. Buchanan [124] demonstrated that following cigarette smoking, the BCG may become abnormal (or more abnormal if the initial record was already abnormal) in patients with clinically evident ischemic heart disease and patients with known coronary disease predisposing conditions, such as diabetes mellitus. Positive BCG smoking tests were also detected in 80% of young patients with thyrotoxicosis but not in patients who were euthyroid following medication or in apparently healthy participants of comparable age. BCG wave anomalies were similar in thyrotoxicosis and coronary disease patients. It is hypothesized that hyperthyroidism’s metabolic demands create an imbalance between oxygen supply and demand, resulting in relative coronary insufficiency.

#### 5.1.1. Sleep Apnea Screening

One-seventh of the world’s population, or close to one billion people, are estimated to have obstructive sleep apnea, the most prevalent kind of sleep-disordered breathing [125]. Almost 45 percent of this population suffer from mild to severe apnea [125]. However, in many cases, it remains undiagnosed [126]. Central sleep apnea is another common sleep-related breathing disorder. While obstructive sleep apnea is due to occluded upper airways, central apnea is caused by increased sensitivity of peripheral and central chemoreceptors, and it is associated with conditions such as heart failure, stroke, and atrial fibrillation [127].

In the long term, these disorders can have deleterious consequences. For example, the exaggerated negative intrathoracic pressure resulting from occluded airways leads to sympathetic nervous system activation and sleep arousals and can cause remodeling of the heart, as well as contribute to the development of cardiovascular diseases [128]. Among others, obstructive sleep apnea can lead to ischemic heart disease, hypertension, and cerebral infarction [129]. Moreover, in patients with left ventricular systolic dysfunction, obstructive sleep apnea can lead to myocardial ischemia and the development of ventricular diastolic or systolic heart diseases [130].

Sleep breathing disorders are often assessed by polysomnography. For this reason, patients are required to attend sleep labs, which is time consuming and costly [131]. In addition to this, it is relatively obtrusive and can disrupt sleep itself [132], even though, ideally, it would be important to acquire data without interfering with or interrupting the subject’s sleep. In view of the high healthcare expenses and the impact of sleep breathing disorders on cardiovascular diseases, an alternative technique that allows home monitoring is required. It has been shown that respiratory and cardiovascular signals can be monitored simultaneously by means of BCG or SCG [117]. This capability in addition to the low-cost and unobtrusive characteristics of these technologies make them suitable alternatives for monitoring vital signs during sleep [82].

So far, it is mainly BCG that has been used in the studies on sleep apnea. As early as in 1986, Salmi et al. [133] presented a static charge sensitive bed that could be used for recording BCG. In this method, the subject lies on a mattress, and various body movements induce a static charge distribution in the active layers [78]. Additionally, the interest in this device for identifying obstructive sleep apnea has been shown in 1989 by Partinen et al. [134]. Later, Lindqvist et al. [135] investigated the connection between esophageal pressure variation and amplitude variation in the static charge sensitive bed BCG. They observed that the amplitude variations in the BCG signal are negatively correlated with the severity of airway obstruction in patients with asthma and chronic obstructive pulmonary disease. They also showed that histamine inhalation, which causes a clinically significant increase in airway obstruction, also increases the amplitude of these respiratory variations.

Additionally, Paalasma et al. [69] developed a bed-based BCG for sleep monitoring at home with a web application, which could perform respiratory variability analysis. In 2015, Zhao et al. [82] and Migliorini et al. [136] used HRV parameters and breathing efforts computed using BCG to detect the sleep apnea syndrome. Subsequently, Liu et al. [137] presented a new framework for enhancing the performance of apneic event detection based on breaking each possible event segment into distinct phases. Very recently, the potential of BCG in sleep screening and sleep apnea detection was further studied [91,132], including with new devices based on microbend fiber optic sensors [138] and with new detection methods, such as convolutional neural networks [84] and wavelet decomposition and the iterative cumulative sums of squares [139].

Guerrero et al. [140] compared the results of a pressure bed sensor and respiratory inductance plethysmography in detecting sleep-related breathing disorders with a reference based on the nasal flow signal. For 25 patients, the correlation coefficient was 0.92 between the pressure bed sensor and the reference signal. Zhao [141] used a mat embedded with a macrobending optical fiber sensor to monitor heart and respiratory rate during sleep. Siyahjani et al. [142] compared a BCG-embedded bed’s performance with polysomnography in terms of estimating epoch-by-epoch heart rate, respiratory rate, sleep vs. wake, and summary sleep variables.

In addition, Alametsa et al. [143] presented a method for detecting spiking events using BCG. Heavy snorers frequently appear with episodes of high-frequency spiking during sleep in addition to apnea or hypopnea [144]. These events are caused by the increased respiratory resistance. These spike episodes have also been linked to partial upper airway blockage [144]. Recognizing spiking events can be useful for assessing the severity of respiratory disturbance during sleep. Kirjavainen et al. [144] studied the static charge sensitive bed capability in measuring breathing stimulation in these events. Their experiments were conducted in awake participants under experimental respiratory challenges, such as free breathing, hypoxia, hypercapnia, and inspiratory and expiratory loading.

The possibility for BCG to simultaneously bring cardiac and respiratory information makes it an interesting technique to evaluate the overall sleep quality. Bader et al. [145] used this technique in patients diagnosed with dementia, as well as in a healthy control group, to evaluate sleep-related respiratory disorders and sleep-related movement disorders. Sadek and Mohktari [146] also assessed sleep quality based on several parameters, such as duration of sleep, sleep interruption, heart rate, and respiration, in three females for over a month in real life. They demonstrated that an optical fiber embedded sensor mat could be used as a novel non-intrusive system for monitoring the quality of sleep. In another study, Zhou et al. [147] presented a material for detection and tracking of dynamic changes in sleep posture in real time, as well as BCG signal monitoring. They also developed a monitoring and intervention system for obstructive sleep apnea in order to improve the sleep quality and prevent sudden death while sleeping. Brink et al. [148] also studied the body movements, respiration, and heart activity of a person at rest or asleep on a bed.

#### 5.1.2. Respiratory Maneuvers

There are several respiratory maneuvers that have been used to investigate cardiorespiratory interactions, including the Valsalva maneuver, the Müller maneuver, and voluntary breath holding.

The Valsalva maneuver is performed by attempting to exhale forcefully against a closed airway [149]. Exhaling against a closed glottis causes changes in intrathoracic pressure that can affect venous return, cardiac output, heart rate, and arterial pressure. The hemodynamic effect of the Valsalva maneuver can be summarized in four distinct phases. In the initial phase, the blood pressure rises in the first few seconds as a result of increased intrathoracic pressure and blood ejection from the left atrium [150]. At the same time, the aortic pressure increases. Then, as the heart rate and aortic pressure have an inverse relationship due to the baroreceptor reflex, the heart rate falls at this phase [150]. In the next phase, due to the increased intrathoracic pressure, the venous return decreases, resulting in a decrease in cardiac output and blood pressure [150]. Then, the aortic pressure begins to fall after a few seconds as the cardiac output decreases [150]. Consequently, the heart rate starts to increase. In phase 3, as the external pressure is released from the aorta, and the subject starts to breathe normally again, there is a brief drop in aortic pressure and a corresponding rise in heart rate as a reflex. There is also an increase in venous return into the right atrium [150]. Finally, phase 4 is characterized by an increase in aortic pressure and a reflex heart rate reduction. In addition, in response to a rapid increase in cardiac filling, the cardiac output abruptly increases [150].

The Müller maneuver is an inspiratory effort against a closed epiglottis. Historically, it was believed to have the opposite effect of the Valsalva maneuver, resulting in an increase in venous return and right ventricular stroke volume [151]. This hypothesis was disproved using both invasive and non-invasive methods. Instead, it was discovered that the Müller maneuver, like Valsalva, decreased the right ventricular stroke volume, which was postulated to be caused by an increase in the impedance of venous return to the right ventricle [150].

Voluntary breath holding is the third maneuver, which can be performed at different phases of the breathing cycle. There are two distinct phases in voluntary breath holding: the initial phase, during which an individual can easily resist the urge to breathe, and the onset of involuntary breathing motions, during which the urge to breathe is no longer suppressed [152]. This phase is traced back to the activation of both central and peripheral chemoreceptors, as well as the deactivation of the autonomic nervous system and the afferent pulmonary wall reflexes. This maneuver places participants under significant dynamic stress because it requires a high level of deliberate control to compensate for the unconscious need to breathe [152].

Several studies suggest that these respiratory maneuvers can be used to diagnose illnesses based on BCG and SCG records. Early on, Onodera [153] performed a study on hypertensive and healthy participants to assess the possibility of using the Valsalva maneuver for the diagnosis of cardiovascular abnormalities. They defined the Valsalva variation index and the respiratory variation index. The former is defined during the Valsalva maneuver, while the latter is defined during rest breathing. The Valsalva variation index was calculated as follows. The IJ wave heights of the three largest wave complexes were averaged. The smallest wave complexes were treated in the same way. The difference between these two values was divided by the first value (the average of the largest IJ waves) and multiplied by 100. The respiratory variation index was calculated based on the method of Brown and de Lalla [49] (see Section 5.1). In this study, Onodera observed a gradual increase in the Valsalva variation index as Brown’s classification grades increased. Based on the Valsalva variation index, they classified BCG into five grades, which were also indicators of the cardiovascular status. Fierro et al. [154] also attempted to show the potential capability of these technologies in hypertension management by collecting data during breath hold and the Valsalva maneuver. They showed that a wearable device equipped with ECG and BCG can be used to predict central aortic blood pressure.

Again, for the purpose of blood pressure monitoring, Shin and Suk Park [94] derived HRV parameters from the BCG signal and statistically compared them to HRV parameters from the ECG signal during rest, the Valsalva maneuver, and static exercise. The Valsalva and static exercise can induce cardiac autonomic rhythm changes, and they demonstrated that the BCG signal can be used to analyze cardiac autonomic modulation. They stated that the RJ interval is correlated with blood pressure. Martin-Yebra et al. [95] further investigated these potential changes in intervals caused by respiratory patterns. They carried out experiments during spontaneous breathing, inspiratory and expiratory breath hold, and Valsalva maneuver in supine and standing positions. Comparing the inspiratory and expiratory breath holds, all temporal parameters, including RR, RI, RJ, RK, and IK, were significantly different between the two conditions in the supine position. In contrast, there were no significant differences in the standing position. In the standing posture, the inspiratory breath hold prolonged all intervals (RK, RJ, IK), except RI, and the Valsalva maneuver prolonged all intervals (RI, RK, RJ), except IK.

The effects of the Valsalva and the Müller maneuvers on BCG were also investigated both by Facci and colleagues [155], and Kazmier and Schild [156,157]. As expected, based on the previous findings, when the intrapulmonary pressure was increased, the amplitude of the BCG decreased; and the opposite was true for decreased intrapulmonary pressure. Bixby [64] came to similar conclusions regarding the Valsalva maneuver [58].

By using the Müller maneuver, Morra et al. [96] revealed that SCG and BCG may be useful for identifying hemodynamic changes during simulated obstructive apneic episodes. Moreover, by conducting voluntary apnea, they demonstrated that kinetic energy parameters based on BCG and SCG can be used to evaluate muscle sympathetic nerve activity changes [75]. Their prior research demonstrated that maximal end expiratory apnea increases these parameters in BCG and SCG [17].

Although the muscular struggle to hold one’s breath for unusual lengths of time may contaminate the records with artifacts, several pioneer studies explored these kinds of experiments. Some of these studies were not originally written in English but were reported and discussed in Starr and Noordergraaf’s book [64]. As a result, they are also mentioned here.

Audier and colleagues [158] studied the effect of breath holding in trained athletes. According to their results, although no ECG changes were found, BCG abnormalities were reported in all but one case at the end of the apnea [64]. In a series of studies on normal subjects, Otis et al. [44] found that while breathing against positive pressure (through a mouthpiece for short periods or a helmet for longer periods), the amplitude of the BCG signal and calculated stroke volume were reduced, while the heart rate increased. When the increase in pressure during breathing was limited to the inspiratory phase, both the stroke volume and heart rate decreased. They concluded that this may be due to the increase in intrapulmonary pressure, which compresses the pulmonary capillaries and thus increases the resistance to pulmonary blood flow. Josenhans [46] also devised studies to distinguish between the effects of changes in intrapulmonary pressure and the effects caused by differences in lung distension. The residual volume and JM wave height were reduced significantly as the intrapulmonary pressure increased at each of the five degrees of inflation they investigated. At constant intrapulmonary pressures, however, an increase in lung distension had little effect on IJ but caused JM to rise significantly.

Douma [159] measured BCG and ECG in subjects who held their breath against a set of pressures. The regression showed that, as the pulmonary pressure increased, the JM wave height diminished, while the ratio of IJ/JM and the QJ interval increased. However, there was a large inter-subject variability regarding the slope of the linear regression and the correlation coefficient. In addition, Polo et al. [129] investigated the correlation of BCG respiratory variations and changes in intrathoracic pressure. They studied healthy subjects during normal breathing, breath holding with constant intrathoracic pressure, and breathing against increased respiratory resistance, and the results indicated that BCG respiratory variations could reflect intrathoracic pressure changes.

Furthermore, by using SCG, Skoric et al. [160] assessed the effect of static respiration volume on the morphology of cardiac-induced waveforms during maximal inspiratory and expiratory breath hold in the SCG linear and rotational signals. They discovered that for high lung volumes, the peak amplitudes of both heart sounds were consistently larger, while the ratio of primary to secondary heart sound was inversely proportional to lung volume.

### 5.2. HRV Analysis

It is recognized that the neural network, particularly the autonomic nervous system, plays a significant role in the interplay between breathing and circulation [24,131]. During inspiration, the decrease in intrathoracic pressure and increase in the abdominal pressure will lead to venous capacitance decrease, increasing longitudinal pressure gradient, and venous return [24]. A drop in intrathoracic pressure can also diminish left ventricular filling, which can cause a reduction in arterial blood pressure. The change in arterial blood pressure can be sensed by aortic and carotid baroreceptors that transmit this information to the brain. Consequently, the autonomic nervous system will decrease vagal activity; thereby, the heart rate will increase during inspiration. The reverse is true during expiration [24].

HRV parameters can be estimated via time-domain, frequency-domain, and non-linear analyses. Conventionally, HRV parameters were extracted from the ECG signal and based on the variations in RR intervals. However, it has been shown that the same can be performed by using only BCG or SCG. In that case, the computations will be performed based on the JJ interval obtained from the BCG or AOAO interval fluctuations from SCG instead of the RR interval from ECG [106,132].

HRV can be quantified by measuring the fluctuations in the intervals of consecutive R, J, or AO peaks. Some of the most important time-domain HRV parameters are the standard deviation of normal-to-normal (NN) interval (SDNN), root mean square of successive differences between intervals (RMSSD), the proportion of the successive pairs of NN intervals that differ more than 50 ms divided by the total number of NN (pNN50) [161].

HRV can also be quantified in the frequency domain, with parameters such as: peak or power in ultra-low frequency (ULF) (≤0.003 Hz), very low frequency (VLF) (0.0033–0.04 Hz), low frequency (LF) (0.04–0.15 Hz), high frequency (HF) (0.15–0.40 Hz), and the ratio between LF and HF (LF/HF) [161]. HF components mainly represent parasympathetic activity. There have been controversial views about LF. LF may be related to both sympathetic and parasympathetic activity. Thus, the ratio LF/HF is established to reflect sympathovagal balance [161].

The non-linear HRV analysis represents the unpredictable and chaotic behaviors of heart rate dynamics [162]. For instance, patients with obstructive sleep apnea have a decreased dynamic complexity [131]. Detrended fluctuation analysis [82], approximate entropy [91], and Poincare plots [88] are other examples of non-linear HRV analyses [162].

#### 5.2.1. BCG

The very first study that used BCG for HRV analysis was conducted in 2010 by Friedrich et al. [163]. They tried to achieve a heart rate estimation method with a beat-to-beat resolution to respond to the needs in the field of sleep and high-risk cardiac pathologies. They used a history-dependent inverse Gaussian process to characterize the stochastic structure of intervals and derived a probability density from it to estimate the RR intervals and HRV [163]. Parchani et al. [164] also computed HRV parameters from BCG and compared them with those derived from ECG. With 54 overnight records from 24 subjects, they reported that the SDNN, RMSSD, and LF/HF computed from BCG were comparable to the corresponding values from ECG. In 2019, Yu et al. [165] also used HRV features from BCG to monitor the stress level of office workers. In 2011, Shin et al. performed HRV analysis using BCG during rest, the Valsalva maneuver, and static exercise in order to evaluate BCG capability of identifying autonomic system activation through its effect on HRV [88].

Based on the link between HRV and sympathovagal activity, Suk Park et al. [89] used BCG-derived HRV parameters for sleep structure estimation. Correspondingly, in 2020, Mitsukura et al. [87] performed sleep stage estimation with HRV parameters from BCG. In moderate to severe sleep apnea, RR variability and the HF component of HRV decrease, while the LF component and LF to HF ratio increase [166]. By using these HRV parameters and based on BCG records, Zhao et al. [82], Gao et al. [132], Migliorini et al. [136] developed methods for sleep apnea detection. The HRV parameters obtained from BCG were further studied by Antink et al. [68]. They performed a study on patients after vascular intervention surgery and healthy subjects to prove the ability of BCG in long-duration monitoring of high-risk patients. Moreover, Liu et al. [167] and Xu et al. [168] revealed that the HRV parameters obtained from BCG could be used as the identifying indicators to detect hypertension and brain fatigue, respectively.

#### 5.2.2. SCG

The first HRV study using SCG was published by Ramos-Castro et al. [103], where they compared the calculated HRV indices from an accelerometer in a smartphone with ECG records. Later, Tadi et al. [106] showed that the estimated HRV indices from SCG and ECG were highly correlated, which again proved the potential of SCG to be used independently and not in the company of ECG. In order to evaluate the validity of HRV indices from SCG, Laurin et al. [107] went one step further and performed this comparison between interbeat estimations from SCG and ECG under an orthostatic stress test. They showed that the HRV results computed from the SCG signal are valid and recommended using the AO waves as a marker to determine heartbeats.

Landreani et al. [169] performed a pilot experiment using mobile phone SCG to evaluate the feasibility of detecting changes in sympathovagal balance by estimating two HRV indices. They demonstrated that the SCG-based changes in HRV between mental rest and mental stress situations were in agreement with ECG and that the RMSSD in particular could be used as a potential marker to measure the stress level. In addition, Hurnanen et al. [108] proposed a method using HRV indices from SCG to identify atrial fibrillation. They reached an average sensitivity and specificity of 99.9% and 96.4% for 13 patients.

### 5.3. Respiratory-Related Features and Classification

The aim of this section is to investigate different types of features that have been used to evaluate respiration-related effects or events. The most common features in BCG and SCG can be categorized as the time domain, frequency domain, and time–frequency domain [36]. However, there are also other innovative approaches.

Typically, the time domain features consist of time intervals and amplitudes of fiducial points, signal energy, HRV time domain features (see Section 5.2), and other statistical features based on them. The frequency features mainly used in this topic are amplitudes and powers in the frequency spectrum and the statistical features related to them, as well as entropy. Finally, the time–frequency features are mainly extracted by wavelet transform. Some demographic parameters, such as age and BMI, can also be included because of their non-negligible impacts on the signal [82,170].

#### 5.3.1. BCG

Although body movements are sometimes one of the most difficult challenges of BCG recordings [138], they have been shown to be effective in detecting sleep apnea. Data from individuals with more severe sleep apnea would have a larger reduction in respiration airflow during sleep apnea episodes, which would result in increased breathing effort [82,91]. In this regard, Zhao et al. [82], in a study on sleep apnea syndrome detection, used wavelet decompositions on the BCG signal to acquire the heartbeat interval; then, they extracted HRV-based characteristics using sample entropy analysis and detrended fluctuation analysis. Breathing effort fluctuation features were also derived using a novel method named the signal turns count. Personal features, such as gender, age, BMI, were also taken into consideration. All the features were employed afterward in the knowledge-based support vector machine classification model.

In another study on obstructive sleep apnea detection, Liu et al. [137] automatically selected the potential event segments from raw BCG data by recognizing arousals. Afterward, an adaptive threshold-based division algorithm was used to divide each possible event segment into three phases (i.e., apnea, respiratory effort, and arousal). At the end, the back propagation neural network was used to classify these possible event segments as either apneic events or non-apneic events. They achieved a precision and sensitivity of 94.6% and 93.1%, respectively. Sadek et al. [138] also investigated the microbend fiber optic sensor’s ability to monitor heart rate and respiration unobtrusively, particularly for sleep apnea patients. By comparing the standard deviation and the mean absolute deviation of the signal’s amplitude in different windows, they were able to divide the data into three categories: body movement, inactivity, and sleep. Then, sleep apnea was detected using an adaptive threshold technique based on the respiratory signal’s standard deviation.

Huysmans et al. [91] obtained the respiratory and BCG signals using prefiltered data, including respiratory signal at [0.08, 3] Hz and BCG signal at [6, 16] Hz. After subtracting the mean value, the prefiltered respiratory signal was bandpass-filtered to [0.08, 2] Hz, then resampled at 4 Hz and 50 Hz, respectively. The signals were normalized afterward because the patient’s weight and position affected the signal amplitude. The time–frequency domain information from the generated signals was extracted using discrete wavelet transform. A robust spectral learning approach was used as an unsupervised feature selection framework. Apneic episodes were then detected using the k-means method.

Cimr et al. [84] designed a convolutional neural network classifier. They used the Cartan curvatures (a geometrical invariant that can be used to evaluate small changes in multivariate time series) and Euclidean arc length of the measured signal from different sensors embedded in the bed for detecting respiratory disorders. Their results showed 96.37% accuracy, 92.46% sensitivity, and 98.11% specificity for 20 healthy subjects. In another study, Ben Nasr et al. [71] classified the BCG signals in phases of normal breathing, coughing, post-coughing, breath hold, expiration, movement, and others by a Gaussian mixture model and K-nearest-neighbor methods. For this purpose, they extracted spectral flatness based on Wiener entropy and spectral centroid.

#### 5.3.2. SCG

In a series of experiments performed during inspiratory and expiratory breath holds, Skoric et al. [160] discovered that the peak amplitudes of both heart sounds were larger for high lung volumes. These respiratory-induced changes can mask other signal variabilities, which may be diagnostically valuable [16,55]. Hence, knowing the inspiratory and expiratory onset or lung volume is critical. The respiratory onsets can be detected in SCG signals by using support vector machine [55,171], K-means clustering [172], principal component analysis, and envelope detection methods [173,174], and again, by using the amplitude and timing of fiducial points [175].

The SCG signal can also be classified in groups of high and low lung volume by support vector machine and radial basis function kernel [176], K-means clustering [172], and adaptive feature extraction method [177]. Furthermore, Taebi and Mansy [109] compared the classification based on flow rate (i.e., inspiration/expiration phase) and lung volume (low/high volume), and it appeared that lung volume rather than flow rate is the major factor in SCG morphological changes, possibly because intrathoracic pressure changes are more correlated with lung volume.

In addition to this, Pandia et al. [110] investigated how respiration affects the frequency domain characteristics of SCG signals. For this purpose, they divided the SCG signal bandwidth into 5 and 10 Hz frequency bins and compared the power in ensemble-averaged inspiratory, expiratory, and apneic beats. By this method, they showed that the power of apneic episodes is significantly different from the inspiratory beats in the frequency range of 10–40 Hz for 20 healthy subjects. Inspiratory and expiratory signals’ powers were also different.

It has also been shown that the respiratory rate can be extracted from SCG by empirical mode decomposition [93,178], wavelet decomposition, or Fourier-based envelope detection [179,180], or by extracting the amplitude and timing features. These features can be S1–S1, and S1–S2 timing intervals, S1 and S2 power or intensity, and S1/S2 ratio [54,178], maximum peaks [97], and amplitude and timing of fiducial points [175]. Ulrich et al. [181] showed that the respiration signal amplitude itself can also be computed from the SCG signal. They used the amplitudes of SCG fiducial points, timings between mitral valve closure and the other fiducial points, mean and median of the frequency component, and the power of the SCG signal to extract respiratory signal amplitude by methods such as multiple regression analysis, support vector regression, and neural network. Moreover, Clairmonte et al. [16] developed a neural network to classify the lung volume state of a subject based on SCG linear and rotational records. Their method could classify the lung volume of 50 subjects with a precision and sensitivity of 99.4% and 99.5%, respectively.

Different levels of respiratory efforts can also be detected by SCG. For instance, Choudhary et al. [182] identified the type of breathing, such as breathlessness, normal breathing, and dyspnea, using SCG and ECG. To achieve this, they employed different features, such as the amplitude of the peak corresponding to aortic valve opening, diastolic energy and entropy, and statistical time domain features. The respiratory effort levels were then detected using a stacked autoencoder neural network.

Azad et al. [183] used the dynamic time warping distances of SCG waveforms in the time domain to show that the variability of SCG in breath hold is less than normal breathing, and Sandler et al. [184] used this feature, as well as the mean and standard deviation of S1 and S2 heart sounds (derived from SCG), for heart failure diagnosis. They detected a shift toward lower frequencies in specific frequency bands in 40 heart failure patients as they approached re-admission. They also showed that these variations in S1 and S2 sounds correspond to the equivalent variations in the heart sound ejection period during regular breathing and breath hold [185].

## 6. Open Research Issues and Future Perspectives

From what has been described up to now, it appears that BCG and SCG can capture cardiac and respiratory information simultaneously, as well as their interaction. While taking respiratory variations into account can improve current applications, such as heartbeat detections methods [186], this also leads to a plethora of additional applications, supported by the relatively wide variety of sensors available. For instance, there have been several pioneer studies on the effect of age [64] and emotions [187] on respiratory variation in BCG records, but these areas of research, just like many applications of BCG and SCG, are still in their infancy. One of the main limiting factors of the recent studies is the generally low number of subjects. Future studies should therefore include larger and more diverse populations. In the past, large-scale studies using BCG have been conducted with very encouraging results [49,120]. However, most of these studies were conducted more than half a century ago and should be reproduced. This is even more critical, as the BCG devices used at this time were generally based on suspended tables [3,4], which is not the case for the current devices (see Table 2). More generally, it is important to compare different modalities to measure BCG and SCG in their abilities to monitor the cardiorespiratory status of a subject.

Among the many applications that would benefit from more studies based on BCG- and SCG-derived cardiorespiratory variability, the case of hypovolemia is particularly interesting. Hypovolemia is defined as a decrease in fluid volume circulating in the body. It can be caused by volume depletion or dehydration [188], resulting from different reasons, such as hemorrhage and orthostatic stress [189]. Long-term exposure to weightlessness is also known to cause hypovolemia [190,191]. The dominant compensatory mechanism in hypovolemia is a reduction in carotid sinus baroreceptor inhibition [192] and changes in the sympathovagal balance [193]. Hypovolemia can be detected by its effect on the respiratory variability of the cardiac stroke volume [194] and on HRV, which, as described in Section 3.3 and Section 5.2, can both be computed by BCG and SCG.

In the context of microgravity, it has been demonstrated that the BCG curve is more stable and less disturbed in microgravity than on the ground under the influence of gravity [195]. In addition, previous studies showed that BCG measurements can be useful for future non-invasive monitoring of the cardiac function, as well as for investigating the effects of respiratory movement on the BCG in three axes [53]. It was also shown that long-term weightlessness affects the respiratory variability of the BCG IJ and JK segments [63]. In particular, in the first month of the mission, the ratio of inspiration to expiration for the IJ segment was less than one (in contrast with terrestrial studies). This was also true for the JK segment for the duration of the 6-month mission. According to the authors, the reason for such changes is that, unlike on earth, the right ventricle volume cannot be increased significantly during inspiration due to the reduced blood volume. Furthermore, because of the increased diastolic pressure in the pulmonary arteries, the velocity of systolic ejection from the right ventricle decreases [63].

A further instance of BCG and SCG applications in extreme environments is to evaluate the consequences of being exposed to hypoxia. Hypoxia can increase systematic blood pressure and activate the sympathetic nervous system [96], and its effect on the cardiopulmonary interaction may be important. To assess the effect of hypoxia on the cardiovascular system, experiments can be conducted during voluntary breath holding [96] on patients with sleep breathing disorders or in hypoxic environments, such as high altitude [196]. For instance, Castiglioni et al. [196] performed a study to evaluate the ability of SCG to identify cardiorespiratory variations during sleep in high-altitude environment compared to the sea level.

Another example of the wide range of possible applications of BCG and SCG is that they can also be helpful in the field of medical imaging. In recent years, Tadi et al. [102] investigated the relationship between SCG respiratory data and the reference respiration belt. According to their findings, there was a significant linear correlation between the accelerometer-based measurement and reference measurement, making SCG a promising option for future use in clinical cardiac positron emission tomography to detect candidate features for cardiac and respiratory gating and, as a result, to obtain motion-free images [197]. Additionally, Nedoma [90,100] proposed a kind of BCG as a solution for continuous monitoring of both respiratory and heart rates inside MRI scanners.

By extension, many applications of HRV should be tested based on BCG and SCG measurements. It is indeed important to evaluate the exact applicability of these unobtrusive and easy to use techniques to serve as a diagnostic or monitoring tool based on the specific HRV parameters. Among the other applications to evaluate, it has been shown that HRV could be used as a diagnostic tool for: diabetes, renal failure, neurological and psychiatric conditions, sleep disorders, psychological phenomena, such as stress, etc. [198].

In addition to this, while these two technologies cannot reach the accuracy level of polysomnography in detecting sleep-disordered breathing, the early results based on BCG and SCG are promising [82,84,91,196]. Despite their potential benefits, there are insufficient data to draw broad conclusions, and more studies in this field are needed. Indeed, the performance of these devices may differ depending on the target group due to comorbidities or age [64]. As a result, it is not yet possible to consider them as stand-alone diagnostic devices. It is also worth noting that some specific applications have yet to be investigated. Among them is the feasibility of using BCG and SCG to evaluate the efficacy of sleep apnea treatments and to measure the impact of sleep apnea on the overall cardiovascular condition. In addition, to the authors’ knowledge, there are not many articles that use SCG to detect sleep-disordered breathing. On the other hand, when used on the International Space Station [199] and at a high altitude [196] for sleep screening, it showed encouraging results. This demonstrates that SCG has the potential to be employed further for this purpose. Moreover, it should be highlighted that the field would benefit from the application of recent state-of-the-art deep-learning techniques for the diagnosis of various pathologies. Among them, in particular, it may prove useful for applications aimed at detecting sleep-disordered breathing, considering the long duration of the records in sleep studies.

Finally, it should be noted that some issues still remain open. One of them is the lack of a standardized approach for evaluating the respiratory variations of BCG and SCG. Moreover, the effect of respiratory variations in different postures should be investigated further. Additionally, as stated before, to gain a clearer image of the respiratory changes of BCG and SCG signals and the link between these variations and different pathologies, more research, ideally with diverse study populations, is required.

## 7. Conclusions

BCG and SCG are helpful techniques for a non-invasive examination of the mechanical functioning of the cardiac and respiratory systems concurrently. The capability of capturing these data using compact, lightweight devices, or even with smartphones, may open up new avenues for the development of home monitoring. Utilizing the same terminology for different devices can aid in reaching better conclusions on the use of these devices and making the most of all current publications. Although processing these data is very challenging because they are prone to artifacts, artificial intelligence can pave the way in this field and help achieve a more accurate and efficient diagnosis. Previously, the main interest in BCG and SCG was to explore the mechanical activity of the heart. However, with recent investigations, the interest in extracting respiratory signals and cardiorespiratory features expands the scope of application and the accuracy of these devices. These features have already been found successful for applications such as measurement of HRV and screening of sleep apnea, among others. In the context of an aging population, this brings hope for addressing the growing need of healthcare. All in all, there is an opportunity for developing the frameworks to define BCG and SCG abnormalities in different grades, as described in the original studies on BCG, as well as determining specific protocols for the diagnosis and follow-up of certain diseases, such as sleep breathing disorders and coronary heart disease.

## Figures and Tables

**Figure 1 sensors-22-09565-f001:**
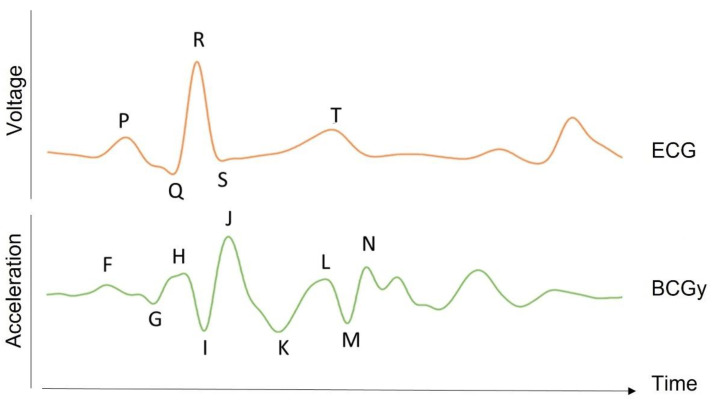
BCG waveform. The time traces of the BCG acceleration signal in the longitudinal axis (y) for a healthy subject are shown together with the ECG signal in one heart beat (in arbitrary unit). As opposed to the SCG, BCG represents global movements, and its waves are more difficult to associate with a single event of the cardiovascular cycle. However, the F, G, and H waves are related to events occurring before the ventricular systolic phase. Indeed, the H wave was shown to be associated with the atrial contraction, while the I and J waves are associated with the ejection of blood in the aorta and other large vessels. The K and following waves are thought to be due to the reflection of the pressure wave at the peripheral vessels and to the resulting oscillations in the center of mass of the overall blood volume.

**Figure 2 sensors-22-09565-f002:**
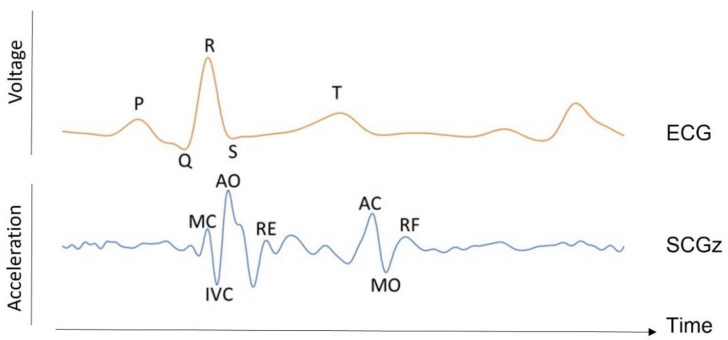
SCG waveform. The time traces of the SCG acceleration signal in the dorsoventral axis (z) for a healthy subject are shown alongside the ECG signal in one heart beat (arbitrary unit). The SCG labels correspond to the mitral valve closure (MC) and opening (MO), aortic valve closure (AC) and opening (AO), rapid ejection (RE), rapid filling (RF), and isovolumetric contraction (IVC).

**Figure 3 sensors-22-09565-f003:**
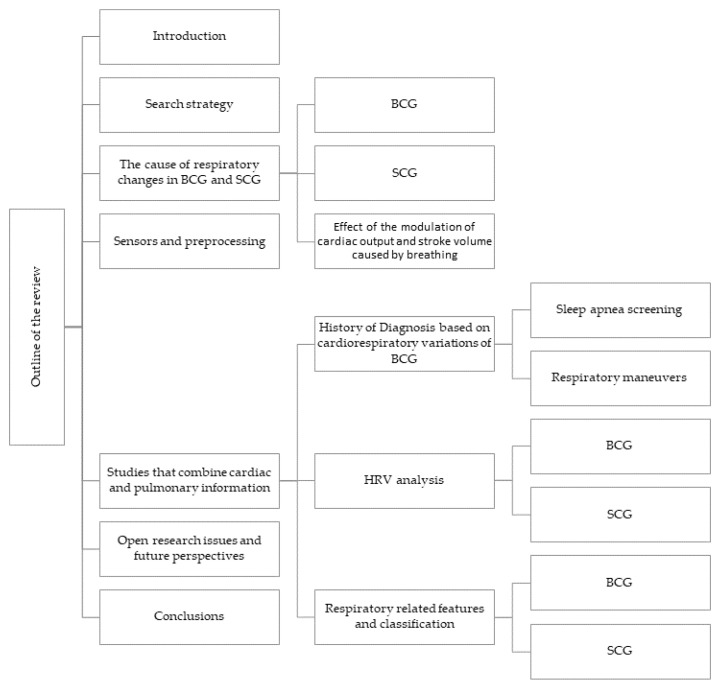
Outline of the review.

**Figure 4 sensors-22-09565-f004:**
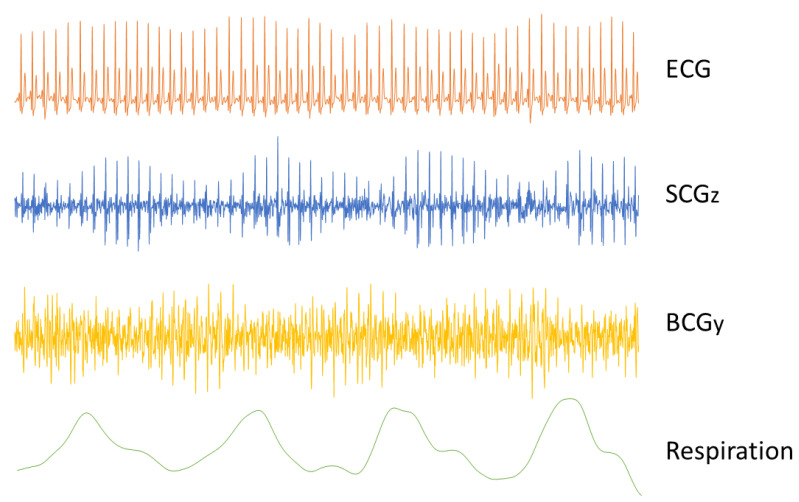
Respiratory variations of the SCG and BCG signals. The time traces of an ECG, SCG (dorsoventral direction), and BCG (head-to-foot direction) are plotted against the concurrently recorded respiration signal. The SCG signal is visibly affected by breathing, with higher amplitudes during expiration than during inspiration. Opposite changes are also visible in the amplitude of the BCG signal, decreasing during expiration.

**Table 1 sensors-22-09565-t001:** Comparison of existing reviews in the field.

Ref.	Year	ECG	PCG	SCG	BCG	Major Contributions
[28]	2011				✓	An overview of research on BCG in microgravity.
[29]	2012				✓	History of BCG, possible future advantages of measuring cardiac cycle events with BCG, and its use in clinical and applied research.
[30]	2012				✓	Review of BCG’s history, its technological advancements up to that point, and recommendations for future studies for home cardiovascular health monitoring.
[31]	2013			✓		Overview of SCG as a non-invasive cardiology method, its historical background, an assessment of the technology at the time, and an appraisal of the issues that should be addressed, such as the development and clarification of definitions, standards, and annotations.
[32]	2015	✓	✓	✓	✓	Review of recent advances in unobtrusive monitoring of cardiorespiratory rhythms, the sensors that have been employed for this purpose, as well as a discussion on identifying the underlying physiological mechanisms, which are observed by different methods.
[8]	2015			✓	✓	Overview of the instrumentation and signal processing advances of BCG and SCG and a summary of the key human subject studies that support the use of BCG and SCG in extra-clinical applications.
[33]	2017			✓		Review focused on extraction of the heart rate using accelerometric technology.
[34]	2018	✓		✓	✓	Overview of unobtrusive monitoring techniques that could be used to monitor some of the human vital signs in a car seat, such as heart rate, breathing rate, temperature, and oxygen saturation.
[35]	2019				✓	Review of BCG sensors and the signal processing methods for analyzing the BCG signal and extracting physiological parameters, such as heart rate and breathing rate.
[36]	2019			✓		Review focused on the developments in the field of SCG, including advances in instrumentation and signal processing.
[37]	2020	✓		✓	✓	Summary of vital function measuring methods integrated into car seats, including ECG, BCG, SCG, signal processing methods and their potential application for the purpose of vital sign monitoring in cars.
[38]	2020			✓	✓	Overview of the current relevance of BCG and SCG for the target medical diagnostics and current research content, such as subject properties or measuring system setups.
[39]	2020			✓		Summary of the history, definition, measurements, waveform description, and applications of gyrocardiography.
[40]	2021			✓	✓	Review on non-invasive atrial fibrillation detection methods based on cardiac dynamics.
[41]	2021	✓			✓	Systematic review of studies conducted on the automated detection of hypertension using ECG, PPG, and BCG signals. Details of the study methods, the physiological signal studied, feature extraction, and diagnostic performance parameters were discussed.
[27]	2021			✓	✓	Current status of research in BCG and SCG applied to the field of medical treatment, health care, and nursing.
[42]	2021	✓		✓	✓	Survey focused on the applications of chest-worn inertial sensors, as well as their placement on the body and the data processing methods associated with each application.
[43]	2022			✓		Review on the currently available wearable devices for measuring precordial vibrations. The focus is on sensor technology and signal processing techniques for the extraction of the desired parameters, applications, and experimental protocols for each technique.

**Table 3 sensors-22-09565-t003:** Summary of SCG sensors’ diversity in the articles discussed.

Scheme	Reference	Sensor Location
Wearable accelerometer	[45,101,102,106,107,108]	sternum
	[109]	left sternal border
	[110]	left mid-sternal area
Smartphone accelerometer	[103,104]	sternum
Radar	[105]	-

## Data Availability

Not applicable.

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
