# Peer review of "Investigating Cardiorespiratory Interaction Using Ballistocardiography and Seismocardiography—A Narrative Review"

_sensors, 2022, doi:10.3390/s22239565_

Round 1

Reviewer 1 Report

Please find the attached file for the reviewers comments. Thanks.

Reviewer 2 Report

Investigating cardiorespiratory interaction using ballistocardiography and seismocardiography – A Review

General comment: some sections have many short paragraphs that read a little disjoined (like a random series of facts).  Better care could be taken to form a clearer narrative

Line 76: " (See Figure 3) "

Comment: Fig 3 is cited in text before Fig 2.  Please relabel Fig 3 to Fig 2, and Fig 2 to Fig 3.

Lines 108-109: "as pre-systolic (F, G)..........and diastolic (L, M, N)."

Comment: F, G, M, N not labelled in figure

Line 137: "Materials and Methods"

Comment: do Review articles have Materials and Methods?

Line 147: "Finally, a total of 182 items relevant to the research were considered."

Comment: is this not Results instead of Methods?

Line 159: "the effect of breathing on the records"

Comment: what records?

Line 317: "a very large respiratory ratio of I wave"

Comment: unsure what this means

Line 500:

Comment: sentence too short and content needs explaining

Line 506: "can be categorized as time-domain, frequency-domain, and time-frequency domain. "

Comment: please add citation(s) here

Line 510: "The main frequency features are amplitudes and powers in the frequency spectrum, and statistical features related to them, as well as entropy. And the time-frequency features mainly can be extracted by wavelet transform. "

Comment: please add citation(s) here

Line 551: "can also be detected by these technologies. "

Comment: what technologies?

Line 12: suggest change "are impacted by the respiration" to "are impacted by respiration"

Line 13 suggest change "which is thus often regarded as an artifact" to "which often produces recording artifacts"

Line 13: suggest change "As a result, either data processing algorithms are developed to exclude the respiratory signals, or the protocol is designed to limit the respiratory bias." to

"As a result, data processing algorithms have been developed to exclude the respiratory signals or recording protocols have been designed to limit the respiratory bias."

Line 19: suggest change "without interfering with the subject's tranquility" to "without introducing recording bias due to irritating monitoring equipment."

Line 25: suggest change "artifacts and relatively" to "artifacts and are relatively"

Line 26: suggest change "intelligence can pave the way in this field and help achieve a more accurate and efficient diagnostic." to "intelligence may help achieve a more accurate and efficient diagnosis."

Line 28: suggest change " of the gold standard methods," to "of the gold standard methods for assessing cardiorespiratory function,"

Line 29: suggest change "the opportunity to perform measurements in real-world conditions," to "the added benefit of being able to perform measurements in real-world situations,"

Line 37: suggest change "and that is used" to "that is used"

Line 39: suggest change "and that is used for determining" to "that is used for measuring"

Line 45: suggest change "to the systematic review" to "to the recent systematic review"

Line 62: suggest change "are changing all along the respiratory cycle" to "are changed depending on when they occur during the respiratory cycle"

Line 65: suggest change "for the changes in heart sound, as Pandia et al. [8] did." to "for the changes in heart sound [8]."

Line 85: suggest change "as reported in [14] it has not yet been demonstrated to provide the timings of SCG fiducial points with reasonable accuracy." to "as reported by Centracchio et al, it has not yet been demonstrated to provide the timings of SCG fiducial points with reasonable accuracy [14]."

Line 87: suggest change "Another technology is gyrocardiography. It " to "Another technology for measuring chest wall vibrations is gyrocardiography, which"

Line 104: suggest change "when Starr [20] started his rigorous examinations [21]" to "after the comprehensive work by Starr [20][21]"

Line 112: suggest change "Early on, in the different studies, it " to "In early studies it"

Line 115: suggest change "BCG records obtained" to "BCG signal"

Line 117: missing "."

Line 139: suggest change "We searched PubMed, IEEEXplore, and Google Scholar for " to "PubMed, IEEEXplore, and Google Scholar were searched for "

Line 157: "This section is mostly focused on BCG early experiments, as it was initiated earlier and studied more extensively. " Could be deleted entirely

Line 168: suggest change "The height of IJ wave" to "The IJ wave height"

Line 171: suggest change "with measurements of stroke volume with ethyl iodide" to "with experimental measurements of stroke volume using ethyl iodide"

Line 181: suggest change "the sum of the IJ and JM amplitudes " to "the sum of the IJ and JM wave heights "

Line 184: suggest change "were" to "included"

Line 189: suggest change "study [25]. Their results showed" to "study [25] which showed"

Line 195: suggest change "believed" to "postulate"

Line 200: suggest change "it was shown by numerical models" to "it was shown in numerical models"

Lines 230, 235: suggest change " JM amplitude" to " JM wave height"?

Line 285: suggest change " increase in IJ wave" to " increase in IJ wave height"

Line 291: suggest change "in coronary heart disease." to "in coronary heart disease [47]."

Line 292: suggest change "BCG waves in different" to "BCG waves during different"

Line 294: suggest change "separating" to "cushioning"

Lines 303, 383: suggest change "IJ amplitude" to "IJ wave height"

Line 342: suggest change " that can be used to further investigate the cardiorespiratory interaction" to " that have been used to investigate cardio-respiratory interactions"

Line 369: suggest change "another maneuver" to "a third maneuver"

Line 370: suggest change "apnea" to "breath holding"

Line 377: suggest change "these maneuvers" to "these respiratory maneuvers"

Line 394: suggest change "for blood pressure monitoring purpose" to "for the purpose of blood pressure monitoring "

Line 423: suggest change "SCG can be" to "SCG has been"

Lines 482, 492: suggest change "signal" to "signals"

Line 622: suggest change "can be" to "has been"

Line 850: suggest change "technologies" to "

Reviewer 3 Report

Dear authors,

thank you for the opportunity to review this article highlighting an interesting topic. There are major points that need adaptation, in particular regarding the structure of the article, as listed below. 

Title:

You should name the article as "A narrative review" directly in the title.

Abstract: 

L. 25: At the end of your abstract you quickly move to future perspectives. At first, you should focus on the results of your study and then move to future perspectives: What is the current state of BCG and SCG? Are currently respiratory signals usually excluded or used for synoptic diagnosis?

Introduction:

L.47: As you performed a narrative review, you should be as concise with your statements as possible, so sentences like "Hence, additional research in this discipline is necessary. " should be removed, as every scientific topic needs further research in general. 

Figure 1: As you compare ECG and SCG data in Figure 1, you should state early in the manuscript, which sensors are needed for the SCG technique and where theyare located on the body. You mention later that there are sternal sensors, but for the reader not familiar with SCG and BCG, an explanation of these techniques should be given right at the beginning.

Figures in general: Please take care that the figures appear chronologically in the manuscript (Figure 3 appears before Figure 2 in the current version). 

L. 124: Please use a corrrect legend for x- and y-axes in all figures. As for SCG, yu should give informations about the sensor placement for BCG right atb the beginning. Furthermore, you should explain why you used the y-axis for BCG and the z-axis for SCG in your figures.

Results

You should add a distinct paragraph to list the pathologies the two methods have been applied to diagnose in the past.

Although generally nicely written, your results part should be better structured. Furthermore, as your "results" are extracts from the existing literature, it does not really differ from the introduction. Thus, the introduction could be shortened extensively. 

I suggest the following structural points for the manuscript in general:

-A strict division between SCG and BCG for all sub-chapters

-Important sub chapters:

1)function of the method

2) use of the method in the literature, history

3) potential diseases that can be or were diagnosed by the methods

4) subdivision in cardial and pulmonal applications

5) Studies that combine cardial and pulmonal informations for both methods seperately. 

Round 2

Reviewer 1 Report

The authors have addressed my previous comments. However, one of the most important section about open research issues in the survey is still missing. The authors mentioned some issues in the abstract but failed to provide a proper section on open research issues before future perspectives section. Few minor revisions are also required. For example,

1. In Table 1 the first column should contain reference number only, there is no need to add authors' names.

2. Add list of acronyms as there are several acronyms in the article.

Reviewer 3 Report

Dear authors, 

thank you for your revised manuscript. Unfortunately, it was submitted in a image based and not text based format, so no search within the manuscript can be performed by the reviewers. Please resubmit in a text based forms.

Based on your Rebuttal letter, there are some comments anyway:

-Don't use BCG and SCG divided by a dash. Please use "and" or "or" to name them both in a proper format.

-Your mentioned adaptation "Line 66: 

The BCG signal can be measured as a displacement, velocity, acceleration, or force signal [8]. It includes movement along all three axes [8]. The initial BCG systems, however, focused on longitudinal (head-to-foot) BCG measurements, which were assumed to represent the largest projection of the 3-D forces induced by cardiac ejection [8].The peaks and valleys in the longitudinal BCG waveform are represented in Error! Reference source not found.. "

is mal formatted in the rebuttal letter and the passage can furthermore not be found in Line 66.

-Figures: You mentioned the meaning for y and z axes now. You should also add the fact that x axis is time.

Round 3

Reviewer 3 Report

Thank you for considering me as a reviewer again. The authors have implanted all adaptations now from my side.

Therefore, the manuscript would be ready for publication from my side.

Best wishes and thank you!